# On the Optimality of Classifier Chain for Multi-label Classification

**Weiwei Liu**          **Ivor W. Tsang**[*]
Centre for Quantum Computation and Intelligent Systems
University of Technology, Sydney
liuweiwei863@gmail.com, ivor.tsang@uts.edu.au

## Abstract

To capture the interdependencies between labels in multi-label classification problems, *classifier chain* (CC) tries to take the multiple labels of each instance into account under a deterministic high-order Markov Chain model. Since its performance is sensitive to the choice of label order, the key issue is how to determine the optimal label order for CC. In this work, we first generalize the CC model over a random label order. Then, we present a theoretical analysis of the generalization error for the proposed generalized model. Based on our results, we propose a *dynamic programming based classifier chain* (CC-DP) algorithm to search the globally optimal label order for CC and a *greedy classifier chain* (CC-Greedy) algorithm to find a locally optimal CC. Comprehensive experiments on a number of real-world multi-label data sets from various domains demonstrate that our proposed CC-DP algorithm outperforms state-of-the-art approaches and the CC-Greedy algorithm achieves comparable prediction performance with CC-DP.

## 1   Introduction

Multi-label classification, where each instance can belong to multiple labels simultaneously, has significantly attracted the attention of researchers as a result of its various applications, ranging from document classification and gene function prediction, to automatic image annotation. For example, a document can be associated with a range of topics, such as *Sports*, *Finance* and *Education* [1]; a gene belongs to the functions of *protein synthesis*, *metabolism* and *transcription* [2]; an image may have both *beach* and *tree* tags [3].

One popular strategy for multi-label classification is to reduce the original problem into many binary classification problems. Many works have followed this strategy. For example, binary relevance (BR) [4] is a simple approach for multi-label learning which independently trains a binary classifier for each label. Recently, Dembczynski *et al.* [5] have shown that methods of multi-label learning which explicitly capture label dependency will usually achieve better prediction performance. Therefore, modeling the label dependency is one of the major challenges in multi-label classification problems. Many multi-label learning models [5, 6, 7, 8, 9, 10, 11, 12] have been developed to capture label dependency. Amongst them, the *classifier chain* (CC) model is one of the most popular methods due to its simplicity and promising experimental results [6].

CC works as follows: One classifier is trained for each label. For the $(i + 1)$th label, each instance is augmented with the 1st, 2nd, $\cdots$, $i$th label as the input to train the $(i + 1)$th classifier. Given a new instance to be classified, CC firstly predicts the value of the first label, then takes this instance together with the predicted value as the input to predict the value of the next label. CC proceeds in this way until the last label is predicted. However, here is the question: *Does the label order affect the performance of CC?* Apparently yes, because different classifier chains involve different

---

[*]Corresponding author

classifiers trained on different training sets. Thus, to reduce the influence of the label order, Read *et al.* [6] proposed the *ensembled classifier chain* (ECC) to average the multi-label predictions of CC over a set of random chain ordering. Since the performance of CC is sensitive to the choice of label order, there is another important question: *Is there any globally optimal classifier chain which can achieve the optimal prediction performance for CC? If yes, how can the globally optimal classifier chain be found?*

To answer the last two questions, we first generalize the CC model over a random label order. We then present a theoretical analysis of the generalization error for the proposed generalized model. Our results show that the upper bound of the generalization error depends on the sum of reciprocal of square of the margin over the labels. Thus, we can answer the second question: the globally optimal CC exists only when the minimization of the upper bound is achieved over this CC. To find the globally optimal CC, we can search over $q!$ different label orders[1], where $q$ denotes the number of labels, which is computationally infeasible for a large $q$. In this paper, we propose the *dynamic programming based classifier chain* (CC-DP) algorithm to simplify the search algorithm, which requires $\mathcal{O}(q^3 n d)$ time complexity. Furthermore, to speed up the training process, a *greedy classifier chain* (CC-Greedy) algorithm is proposed to find a locally optimal CC, where the time complexity of the CC-Greedy algorithm is $\mathcal{O}(q^2 n d)$.

**Notations:** Assume $\mathbf{x}_t \in \mathbb{R}^d$ is a real vector representing an input or instance (feature) for $t \in \{1, \cdots, n\}$. $n$ denotes the number of training samples. $\mathcal{Y}_t \subseteq \{\lambda_1, \lambda_2, \cdots, \lambda_q\}$ is the corresponding output (label). $\mathbf{y}_t \in \{0, 1\}^q$ is used to represent the label set $\mathcal{Y}_t$, where $\mathbf{y}_t(j) = 1$ if and only if $\lambda_j \in \mathcal{Y}_t$.

# 2   Related work and preliminaries

To capture label dependency, Hsu *et al.* [13] first use compressed sensing technique to handle the multi-label classification problem. They project the original label space into a low dimensional label space. A regression model is trained on each transformed label. Recovering multi-labels from the regression output usually involves solving a quadratic programming problem [13], and many works have been developed in this way [7, 14, 15]. Such methods mainly aim to use different projection methods to transform the original label space into another effective label space.

Another important approach attempts to exploit the different orders (first-order, second-order and high-order) of label correlations [16]. Following this way, some works also try to provide a probabilistic interpretation for label correlations. For example, Guo and Gu [8] model the label correlations using a conditional dependency network; PCC [5] exploits a high-order Markov Chain model to capture the correlations between the labels and provide an accurate probabilistic interpretation of CC. Other works [6, 9, 10] focus on modeling the label correlations in a deterministic way, and CC is one of the most popular methods among them. This work will mainly focus on the deterministic high-order classifier chain.

## 2.1   Classifier chain

Similar to BR, the *classifier chain* (CC) model [6] trains $q$ binary classifiers $h_j$ ($j \in \{1, \cdots, q\}$). Classifiers are linked along a chain where each classifier $h_j$ deals with the binary classification problem for label $\lambda_j$. The augmented vector $\{\mathbf{x}_t, \mathbf{y}_t(1), \cdots, \mathbf{y}_t(j)\}_{t=1}^n$ is used as the input for training classifier $h_{j+1}$. Given a new testing instance $\mathbf{x}$, classifier $h_1$ in the chain is responsible for predicting the value of $\mathbf{y}(1)$ using input $\mathbf{x}$. Then, $h_2$ predicts the value of $\mathbf{y}(2)$ taking $\mathbf{x}$ plus the predicted value of $\mathbf{y}(1)$ as an input. Following in this way, $h_{j+1}$ predicts $\mathbf{y}(j + 1)$ using the predicted value of $\mathbf{y}(1), \cdots, \mathbf{y}(j)$ as additional input information. CC passes label information between classifiers, allowing CC to exploit the label dependence and thus overcome the label independence problem of BR. Essentially, it builds a deterministic high-order Markov Chain model to capture the label correlations.

## 2.2  Ensembled classifier chain

Different classifier chains involve different classifiers learned on different training sets and thus the order of the chain itself clearly affects the prediction performance. To solve the issue of selecting a chain order for CC, Read *et al.* [6] proposed the extension of CC, called *ensembled classifier chain* (ECC), to average the multi-label predictions of CC over a set of random chain ordering. ECC first randomly reorders the labels $\{\lambda_1, \lambda_2, \cdots, \lambda_q\}$ many times. Then, CC is applied to the reordered labels for each time and the performance of CC is averaged over those times to obtain the final prediction performance.

# 3  Proposed model and generalization error analysis

## 3.1  Generalized classifier chain

We generalize the CC model over a random label order, called *generalized classifier chain* (GCC) model. Assume the labels $\{\lambda_1, \lambda_2, \cdots, \lambda_q\}$ are randomly reordered as $\{\zeta_1, \zeta_2, \cdots, \zeta_q\}$, where $\zeta_j = \lambda_k$ means label $\lambda_k$ moves to position $j$ from $k$. In the GCC model, classifiers are also linked along a chain where each classifier $h_j$ deals with the binary classification problem for label $\zeta_j$ ($\lambda_k$). GCC follows the same training and testing procedures as CC, while the only difference is the label order. In the GCC model, for input $\mathbf{x}_t$, $\mathbf{y}_t(j) = 1$ if and only if $\zeta_j \in \mathcal{Y}_t$.

## 3.2  Generalization error analysis

In this section, we analyze the generalization error bound of the multi-label classification problem using GCC based on the techniques developed for the generalization performance of classifiers with a large margin [17] and perceptron decision tree [18].

Let $\mathbf{X}$ represent the input space. Both $\mathbf{s}$ and $\bar{\mathbf{s}}$ are $m$ samples drawn independently according to an unknown distribution $D$. We denote logarithms to base 2 by $\log$. If $\mathcal{S}$ is a set, $|\mathcal{S}|$ denotes its cardinality. $\|\cdot\|$ means the $l_2$ norm. We train a support vector machine(SVM) for each label $\zeta_j$. Let $\{\mathbf{x}_t\}_{t=1}^n$ as the feature and $\{\mathbf{y}_t(\zeta_j)\}_{t=1}^n$ as the label, the output parameter of SVM is defined as $[\mathbf{w}_j, b_j] = SVM(\{\mathbf{x}_t, \mathbf{y}_t(\zeta_1), \cdots, \mathbf{y}_t(\zeta_{j-1})\}_{t=1}^n, \{\mathbf{y}_t(\zeta_j)\}_{t=1}^n)$. The margin for label $\zeta_j$ is defined as:

$$\gamma^j = \frac{1}{||\mathbf{w}_j||^2} \tag{1}$$

We begin with the definition of the fat shattering dimension.

**Definition 1** ([19]). *Let $\mathcal{H}$ be a set of real valued functions. We say that a set of points $P$ is $\gamma$-shattered by $\mathcal{H}$ relative to $r = (r_p)_{p \in P}$ if there are real numbers $r_p$ indexed by $p \in P$ such that for all binary vectors $b$ indexed by $P$, there is a function $f_b \in \mathcal{H}$ satisfying*

$$f_b(p) = \begin{cases} \geq r_p + \gamma & \text{if } b_p = 1 \\ \leq r_p - \gamma & \text{otherwise} \end{cases}$$

*The fat shattering dimension $fat(\gamma)$ of the set $\mathcal{H}$ is a function from the positive real numbers to the integers which maps a value $\gamma$ to the size of the largest $\gamma$-shattered set, if this is finite, or infinity otherwise.*

Assume $\mathcal{H}$ is the real valued function class and $h \in \mathcal{H}$. $l(y, h(x))$ denotes the loss function. The expected error of $h$ is defined as $er_D[h] = E_{(x,y) \sim D}[l(y, h(x))]$, where $(x, y)$ drawn from the unknown distribution $D$. Here we select 0-1 loss function. So, $er_D[h] = P_{(x,y) \sim D}(h(x) \neq y)$. $er_{\mathbf{s}}[h]$ is defined as $er_{\mathbf{s}}[h] = \frac{1}{n} \sum_{t=1}^{n} [\![y_t \neq h(x_t)]\!].^2$

Suppose $\mathcal{N}(\epsilon, \mathcal{H}, \mathbf{s})$ is the $\epsilon$-covering number of $\mathcal{H}$ with respect to the $l_\infty$ pseudo-metric measuring the maximum discrepancy on the sample $\mathbf{s}$. The notion of the covering number can be referred to the Supplementary Materials. We introduce the following general corollary regarding the bound of the covering number:

**Corollary 1** ([17]). *Let $\mathcal{H}$ be a class of functions $X \to [a, b]$ and $D$ a distribution over $X$. Choose $0 < \epsilon < 1$ and let $d = fat(\epsilon/4) \le em$. Then*

$$E(\mathcal{N}(\epsilon, \mathcal{H}, \boldsymbol{s})) \le 2\Big(\frac{4m(b-a)^2}{\epsilon^2}\Big)^{d \log(2em(b-a)/(d\epsilon))} \tag{2}$$

*where the expectation $E$ is over samples $\boldsymbol{s} \in X^m$ drawn according to $D^m$.*

We study the generalization error bound of the specified GCC with the specified number of labels and margins. Let $G$ be the set of classifiers of GCC, $G = \{h_1, h_2, \cdots, h_q\}$. $er_{\boldsymbol{s}}[G]$ denotes the fraction of the number of errors that GCC makes on $\boldsymbol{s}$. Define $\hat{\mathbf{x}} \in \mathbf{X} \times \{0, 1\}$, $\hat{h}_j(\hat{\mathbf{x}}) = h_j(\mathbf{x})(1 - \mathbf{y}(j)) - h_j(\mathbf{x})\mathbf{y}(j)$. If an instance $\mathbf{x} \in \mathbf{X}$ is correctly classified by $h_j$, then $\hat{h}_j(\hat{\mathbf{x}}) < 0$. Moreover, we introduce the following proposition:

**Proposition 1.** *If an instance $\boldsymbol{x} \in \boldsymbol{X}$ is misclassified by a GCC model, then $\exists h_j \in G, \hat{h}_j(\hat{\mathbf{x}}) \ge 0$.*

**Lemma 1.** *Given a specified GCC model with $q$ labels and with margins $\gamma^1, \gamma^2, \cdots, \gamma^q$ for each label satisfying $k_i = fat(\gamma^i/8)$, where $fat$ is continuous from the right. If GCC has correctly classified $m$ multi-labeled examples $\boldsymbol{s}$ generated independently according to the unknown (but fixed) distribution $D$ and $\bar{\boldsymbol{s}}$ is a set of another $m$ multi-labeled examples, then we can bound the following probability to be less than $\delta$: $P^{2m}\{\boldsymbol{s}\bar{\boldsymbol{s}} : \exists \, a \, GCC \, model, \, it \, correctly \, classifies \, \boldsymbol{s}, \, fraction \, of \, \bar{\boldsymbol{s}} \, misclassified > \epsilon(m, q, \delta)\} < \delta$, where $\epsilon(m, q, \delta) = \frac{1}{m}(Q \log(32m) + \log \frac{2^q}{\delta})$ and $Q = \sum_{i=1}^{q} k_i \log(\frac{8em}{k_i})$.*

*Proof.* (of Lemma 1). Suppose $G$ is a GCC model with $q$ labels and with margins $\gamma^1, \gamma^2, \cdots, \gamma^q$, the probability event in Lemma 1 can be described as

$$A = \{\boldsymbol{s}\bar{\boldsymbol{s}} : \exists G, k_i = fat(\gamma^i/8), er_{\boldsymbol{s}}[G] = 0, er_{\bar{\boldsymbol{s}}}[G] > \epsilon\}.$$

Let $\hat{\boldsymbol{s}}$ and $\hat{\bar{\boldsymbol{s}}}$ denote two different set of $m$ examples, which are drawn i.i.d. from the distribution $D \times \{0, 1\}$. Applying the definition of $\hat{\mathbf{x}}$, $\hat{h}$ and Proposition 1, the event can also be written as $A = \{\hat{\boldsymbol{s}}\hat{\bar{\boldsymbol{s}}} : \exists G, \hat{\gamma}^i = \gamma^i/2, k_i = fat(\hat{\gamma}^i/4), er_{\boldsymbol{s}}[G] = 0, r_i = max_t\hat{h}_i(\hat{\mathbf{x}}_t), 2\hat{\gamma}^i = -r_i, |\{\hat{\mathbf{y}} \in \hat{\bar{\boldsymbol{s}}} : \exists h_i \in G, \hat{h}_i(\hat{\mathbf{y}}) \ge 2\hat{\gamma}^i + r_i\}| > m\epsilon\}$. Here, $-max_t\hat{h}_i(\hat{\mathbf{x}}_t)$ means the minimal value of $|h_i(\mathbf{x})|$ which represents the margin for label $\zeta_i$, so $2\hat{\gamma}^i = -r_i$. Let $\gamma_{k_i} = min\{\gamma' : fat(\gamma'/4) \le k_i\}$, so $\gamma_{k_i} \le \hat{\gamma}^i$, we define the following function:

$$\pi(\hat{h}) = \begin{cases} 0 & \text{if } \hat{h} \ge 0 \\ -2\gamma_{k_i} & \text{if } \hat{h} \le -2\gamma_{k_i} \\ \hat{h} & \text{otherwise} \end{cases}$$

so $\pi(\hat{h}) \in [-2\gamma_{k_i}, 0]$. Let $\pi(\hat{G}) = \{\pi(\hat{h}) : h \in G\}$.

Let $B_{\hat{\boldsymbol{s}}\hat{\bar{\boldsymbol{s}}}}^{k_i}$ represent the minimal $\gamma_{k_i}$-cover set of $\pi(\hat{G})$ in the pseudo-metric $d_{\hat{\boldsymbol{s}}\hat{\bar{\boldsymbol{s}}}}$. We have that for any $h_i \in G$, there exists $\tilde{f} \in B_{\hat{\boldsymbol{s}}\hat{\bar{\boldsymbol{s}}}}^{k_i}$, $|\pi(\hat{h}_i(\hat{\mathbf{z}})) - \pi(\tilde{f}(\hat{\mathbf{z}}))| < \gamma_{k_i}$, for all $\hat{\mathbf{z}} \in \hat{\boldsymbol{s}}\hat{\bar{\boldsymbol{s}}}$. For all $\hat{\mathbf{x}} \in \hat{\boldsymbol{s}}$, by the definition of $r_i$, $\hat{h}_i(\hat{\mathbf{x}}) \le r_i = -2\hat{\gamma}^i$, and $\gamma_{k_i} \le \hat{\gamma}^i$, $\hat{h}_i(\hat{\mathbf{x}}) \le -2\gamma_{k_i}$, $\pi(\hat{h}_i(\hat{\mathbf{x}})) = -2\gamma_{k_i}$, so $\pi(\tilde{f}(\hat{\mathbf{x}})) < -2\gamma_{k_i} + \gamma_{k_i} = -\gamma_{k_i}$. However, there are at least $m\epsilon$ points $\hat{\mathbf{y}} \in \hat{\bar{\boldsymbol{s}}}$ such that $\hat{h}_i(\hat{\mathbf{y}}) \ge 0$, so $\pi(\tilde{f}(\hat{\mathbf{y}})) > -\gamma_{k_i} > max_t\pi(\tilde{f}(\hat{\mathbf{x}}_t))$. Since $\pi$ only reduces separation between output values, we conclude that the inequality $\tilde{f}(\hat{\mathbf{y}}) > max_t\tilde{f}(\hat{\mathbf{x}}_t)$ holds. Moreover, the $m\epsilon$ points in $\hat{\bar{\boldsymbol{s}}}$ with the largest $\tilde{f}$ values must remain for the inequality to hold. By the permutation argument, at most $2^{-m\epsilon}$ of the sequences obtained by swapping corresponding points satisfy the conditions for fixed $\tilde{f}$.

As for any $h_i \in G$, there exists $\tilde{f} \in B_{\hat{\boldsymbol{s}}\hat{\bar{\boldsymbol{s}}}}^{k_i}$, so there are $|B_{\hat{\boldsymbol{s}}\hat{\bar{\boldsymbol{s}}}}^{k_i}|$ possibilities of $\tilde{f}$ that satisfy the inequality for $k_i$. Note that $|B_{\hat{\boldsymbol{s}}\hat{\bar{\boldsymbol{s}}}}^{k_i}|$ is a positive integer which is usually bigger than 1 and by the union bound, we get the following inequality:

$$P(A) \le (E(|B_{\hat{\boldsymbol{s}}\hat{\bar{\boldsymbol{s}}}}^{k_1}|) + \cdots + E(|B_{\hat{\boldsymbol{s}}\hat{\bar{\boldsymbol{s}}}}^{k_q}|))2^{-m\epsilon} \le (E(|B_{\hat{\boldsymbol{s}}\hat{\bar{\boldsymbol{s}}}}^{k_1}|) \times \cdots \times E(|B_{\hat{\boldsymbol{s}}\hat{\bar{\boldsymbol{s}}}}^{k_q}|))2^{-m\epsilon}$$

Since every set of points $\gamma$-shattered by $\pi(\hat{G})$ can be $\gamma$-shattered by $\hat{G}$, so $fat_{\pi(\hat{G})}(\gamma) \le fat_{\hat{G}}(\gamma)$, where $\hat{G} = \{\hat{h} : h \in G\}$. Hence, by Corollary 1 (setting $[a, b]$ to $[-2\gamma_{k_i}, 0]$, $\epsilon$ to $\gamma_{k_i}$ and $m$ to $2m$),

$$E(|B_{\hat{\boldsymbol{s}}\hat{\bar{\boldsymbol{s}}}}^{k_i}|) = E(\mathcal{N}(\gamma_{k_i}, \pi(\hat{G}), \hat{\boldsymbol{s}}\hat{\bar{\boldsymbol{s}}})) \le 2(32m)^{d \log(\frac{8em}{d})}$$

where $d = fat_{\pi(\hat{G})}(\gamma_{k_i}/4) \leq fat_{\hat{G}}(\gamma_{k_i}/4) \leq k_i$. Thus $E(|B_{\hat{\mathbf{s}}\hat{\mathbf{s}}}^{k_i}|) \leq 2(32m)^{k_i \log(\frac{8em}{k_i})}$, and we obtain

$$P(A) \leq (E(|B_{\hat{\mathbf{s}}\hat{\mathbf{s}}}^{k_1}|) \times \cdots \times E(|B_{\hat{\mathbf{s}}\hat{\mathbf{s}}}^{k_q}|))2^{-m\epsilon} \leq \prod_{i=1}^{q} 2(32m)^{k_i \log(\frac{8em}{k_i})} = 2^q(32m)^Q$$

where $Q = \sum_{i=1}^{q} k_i \log(\frac{8em}{k_i})$. And so $(E(|B_{\hat{\mathbf{s}}\hat{\mathbf{s}}}^{k_1}|) \times \cdots \times E(|B_{\hat{\mathbf{s}}\hat{\mathbf{s}}}^{k_q}|))2^{-m\epsilon} < \delta$ provided

$$\epsilon(m, q, \delta) \geq \frac{1}{m}\Big(Q \log(32m) + \log \frac{2^q}{\delta}\Big)$$

as required. $\qquad\qquad\qquad\qquad\qquad\qquad\qquad\qquad\qquad\qquad\qquad\qquad\qquad\qquad\qquad\square$

Lemma 1 applies to a particular GCC model with a specified number of labels and a specified margin for each label. In practice, we will observe the margins after running the GCC model. Thus, we must bound the probabilities uniformly over all of the possible margins that can arise to obtain a practical bound. The generalization error bound of the multi-label classification problem using GCC is shown as follows:

**Theorem 1.** *Suppose a random $m$ multi-labeled sample can be correctly classified using a GCC model, and suppose this GCC model contains $q$ classifiers with margins $\gamma^1, \gamma^2, \cdots, \gamma^q$ for each label. Then we can bound the generalization error with probability greater than $1 - \delta$ to be less than*

$$\frac{130R^2}{m}\Big(Q' \log(8em) \log(32m) + \log \frac{2(2m)^q}{\delta}\Big)$$

*where $Q' = \sum_{i=1}^{q} \frac{1}{(\gamma^i)^2}$ and $R$ is the radius of a ball containing the support of the distribution.*

Before proving Theorem 1, we state one key Symmetrization lemma and Theorem 2.

**Lemma 2** (Symmetrization). *Let $\mathcal{H}$ be the real valued function class. $\mathbf{s}$ and $\bar{\mathbf{s}}$ are $m$ samples both drawn independently according to the unknown distribution $D$. If $m\epsilon^2 \geq 2$, then*

$$P_{\mathbf{s}}(\sup_{h \in \mathcal{H}} |er_D[h] - er_{\mathbf{s}}[h]| \geq \epsilon) \leq 2P_{\mathbf{s}\bar{\mathbf{s}}}(\sup_{h \in \mathcal{H}} |er_{\bar{\mathbf{s}}}[h] - er_{\mathbf{s}}[h]| \geq \epsilon/2) \qquad (3)$$

The proof details of this lemma can be found in the Supplementary Material.

**Theorem 2** ([20]). *Let $\mathcal{H}$ be restricted to points in a ball of $\mathcal{M}$ dimensions of radius $R$ about the origin, then*

$$fat_{\mathcal{H}}(\gamma) \leq \min\Big\{\frac{R^2}{\gamma^2}, \mathcal{M} + 1\Big\} \qquad (4)$$

*Proof.* (of Theorem 1). We must bound the probabilities over different margins. We first use Lemma 2 to bound the probability of error in terms of the probability of the discrepancy between the performance on two halves of a double sample. Then we combine this result with Lemma 1. We must consider all possible patterns of $k_i$'s for label $\zeta_i$. The largest value of $k_i$ is $m$. Thus, for fixed $q$, we can bound the number of possibilities by $m^q$. Hence, there are $m^q$ of applications of Lemma 1.

Let $c_i = \{\gamma^1, \gamma^2, \cdots, \gamma^q\}$ denote the $i$-th combination of margins varied in $\{1, \cdots, m\}^q$. $\mathcal{G}$ denotes a set of GCC models. The generalization error of $G$ can be represented as $er_D[G]$ and $er_{\mathbf{s}}[G]$ is 0, where $G \in \mathcal{G}$. The uniform convergence bound of the generalization error is

$$P_{\mathbf{s}}(\sup_{G \in \mathcal{G}} |er_D[G] - er_{\mathbf{s}}[G]| \geq \epsilon)$$

Applying Lemma 2,

$$P_{\mathbf{s}}(\sup_{G \in \mathcal{G}} |er_D[G] - er_{\mathbf{s}}[G]| \geq \epsilon) \leq 2P_{\mathbf{s}\bar{\mathbf{s}}}(\sup_{G \in \mathcal{G}} |er_{\bar{\mathbf{s}}}[G] - er_{\mathbf{s}}[G]| \geq \epsilon/2)$$

Let $J_{c_i} = \{\mathbf{s}\bar{\mathbf{s}} : \exists$ a GCC model $G$ with $q$ labels and with margins $c_i : k_i = fat(\gamma^i/8), er_{\mathbf{s}}[G] = 0, er_{\bar{\mathbf{s}}}[G] \geq \epsilon/2\}$. Clearly,

$$P_{\mathbf{s}\bar{\mathbf{s}}}(\sup_{G \in \mathcal{G}} |er_{\bar{\mathbf{s}}}[G] - er_{\mathbf{s}}[G]| \geq \epsilon/2) \leq P^{m^q}\Big(\bigcup_{i=1}^{m^q} J_{c_i}\Big)$$

As $k_i$ still satisfies $k_i = fat(\gamma^i/8)$, Lemma 1 can still be applied to each case of $P^{m^q}(J_{c_i})$. Let $\delta_k = \delta/m^q$. Applying Lemma 1 (replacing $\delta$ by $\delta_k/2$), we get:

$$P^{m^q}(J_{c_i}) < \delta_k/2$$

where $\epsilon(m, k, \delta_k/2) \geq 2/m(Q \log(32m) + \log \frac{2 \times 2^q}{\delta_k})$ and $Q = \sum_{i=1}^{q} k_i \log(\frac{4em}{k_i})$. By the union bound, it suffices to show that $P^{m^q}(\bigcup_{i=1}^{m^q} J_{c_i}) \leq \sum_{i=1}^{m^q} P^{m^q}(J_{c_i}) < \delta_k/2 \times m^q = \delta/2$. Applying Lemma 2,

$$P_{\mathbf{s}}(\sup_{G \in \mathcal{G}} |er_D[G] - er_{\mathbf{s}}[G]| \geq \epsilon) \leq 2P_{\mathbf{s\bar{s}}}(\sup_{G \in \mathcal{G}} |er_{\bar{\mathbf{s}}}[G] - er_{\mathbf{s}}[G]| \geq \epsilon/2)$$

$$\leq 2P^{m^q}\Big(\bigcup_{i=1}^{m^q} J_{c_i}\Big) < \delta$$

Thus, $P_{\mathbf{s}}(\sup_{G \in \mathcal{G}} |er_D[G] - er_{\mathbf{s}}[G]| \leq \epsilon) \geq 1 - \delta$. Let $R$ be the radius of a ball containing the support of the distribution. Applying Theorem 2, we get $k_i = fat(\gamma^i/8) \leq 65R^2/(\gamma^i)^2$. Note that we have replaced the constant $8^2 = 64$ by 65 in order to ensure the continuity from the right required for the application of Lemma 1. We have upperbounded $\log(8em/k_i)$ by $\log(8em)$. Thus,

$$er_D[G] \leq 2/m\Big(Q \log(32m) + \log \frac{2(2m)^q}{\delta}\Big)$$

$$\leq \frac{130R^2}{m}\Big(Q' \log(8em) \log(32m) + \log \frac{2(2m)^q}{\delta}\Big)$$

where $Q' = \sum_{i=1}^{q} \frac{1}{(\gamma^i)^2}$. □

Given the training data size and the number of labels, Theorem 1 reveals one important factor in reducing the generalization error bound for the GCC model: the minimization of the sum of reciprocal of square of the margin over the labels. Thus, we obtain the following Corollary:

**Corollary 2** (Globally Optimal Classifier Chain). *Suppose a random $m$ multi-labeled sample with $q$ labels can be correctly classified using a GCC model, this GCC model is the globally optimal classifier chain if and only if the minimization of $Q'$ in Theorem 1 is achieved over this classifier chain.*

Given the number of labels $q$, there are $q!$ different label orders. It is very expensive to find the globally optimal CC, which can minimize $Q'$, by searching over all of the label orders. Next, we discuss two simple algorithms.

## 4 Optimal classifier chain algorithm

In this section, we propose two simple algorithms for finding the optimal CC based on our result in Section 3. To clearly state the algorithms, we redefine the margins with label order information. Given label set $\mathcal{M} = \{\lambda_1, \lambda_2, \cdots, \lambda_q\}$, suppose a GCC model contains $q$ classifiers. Let $o_i(1 \leq o_i \leq q)$ denote the order of $\lambda_i$ in the GCC model, $\gamma_i^{o_i}$ represents the margin for label $\lambda_i$, with previous $o_i - 1$ labels as the augmented input. If $o_i = 1$, then $\gamma_i^1$ represents the margin for label $\lambda_i$, without augmented input. Then $Q'$ is redefined as $Q' = \sum_{i=1}^{q} \frac{1}{(\gamma_i^{o_i})^2}$.

### 4.1 Dynamic programming algorithm

To simplify the search algorithm mentioned before, we propose the CC-DP algorithm to find the globally optimal CC. Note that $Q' = \sum_{i=1}^{q} \frac{1}{(\gamma_i^{o_i})^2} = \frac{1}{(\gamma_q^{o_q})^2} + \cdots + \big[\frac{1}{(\gamma_{k+1}^{o_{k+1}})^2} + \sum_{j=1}^{k} \frac{1}{(\gamma_j^{o_j})^2}\big]$, we explore the idea of DP to iteratively optimize $Q'$ over a subset of $\mathcal{M}$ with the length of $1, 2, \cdots, q$. Finally, we can obtain the optimal $Q'$ over $\mathcal{M}$. Assume $i \in \{1, \cdots, q\}$. Let $V(i, \eta)$ be the optimal $Q'$ over a subset of $\mathcal{M}$ with the length of $\eta(1 \leq \eta \leq q)$, where the label order is ending by label $\lambda_i$. Suppose $M_i^\eta$ represent the corresponding label set for $V(i, \eta)$. When $\eta = q$, $V(i, q)$ be the optimal $Q'$ over $\mathcal{M}$, where the label order is ending by label $\lambda_i$. The DP equation is written as:

$$V(i, \eta + 1) = \min_{j \neq i, \lambda_i \notin M_j^\eta} \left\{ \frac{1}{(\gamma_i^{\eta+1})^2} + V(j, \eta) \right\} \tag{5}$$

where $\gamma_i^{\eta+1}$ is the margin for label $\lambda_i$, with $M_j^\eta$ as the augmented input. The initial condition of DP is: $V(i,1) = \frac{1}{(\gamma_i^1)^2}$ and $M_i^1 = \{\lambda_i\}$. Then, the optimal $Q'$ over $\mathcal{M}$ can be obtained by solving $\min_{i\in\{1,\cdots,q\}} V(i,q)$. Assume the training of linear SVM takes $\mathcal{O}(nd)$. The CC-DP algorithm is shown as the following bottom-up procedure: from the bottom, we first compute $V(i,1) = \frac{1}{(\gamma_i^1)^2}$, which takes $\mathcal{O}(nd)$. Then we compute $V(i,2) = \min_{j\neq i, \lambda_i\notin M_j^1}\{\frac{1}{(\gamma_i^2)^2} + V(j,1)\}$, which requires at most $\mathcal{O}(qnd)$, and set $M_i^2 = M_j^1 \cup \{\lambda_i\}$. Similarly, it takes at most $\mathcal{O}(q^2nd)$ time complexity to calculate $V(i,q)$. Last, we iteratively solve this DP Equation, and use $\min_{i\in\{1,\cdots,q\}} V(i,q)$ to get the optimal solution, which requires at most $\mathcal{O}(q^3nd)$ time complexity.

**Theorem 3** (Correctness of CC-DP). *$Q'$ can be minimized by CC-DP, which means this Algorithm can find the globally optimal CC.*

The proof can be referred to in the Supplementary Materials.

## 4.2 Greedy algorithm

We propose a CC-Greedy algorithm to find a locally optimal CC to speed up the CC-DP algorithm. To save time, we construct only one classifier chain with the locally optimal label order. Based on the training instances, we select the label from $\{\lambda_1, \lambda_2, \cdots, \lambda_q\}$ as the first label, if the maximum margin can be achieved over this label, without augmented input. The first label is denoted by $\zeta_1$. Then we select the label from the remainder as the second label, if the maximum margin can be achieved over this label with $\zeta_1$ as the augmented input. We continue in this way until the last label is selected. Finally, this algorithm will converge to the locally optimal CC. We present the details of the CC-Greedy algorithm in the Supplementary Materials, where the time complexity of this algorithm is $\mathcal{O}(q^2nd)$.

# 5 Experiment

In this section, we perform experimental studies on a number of benchmark data sets from different domains to evaluate the performance of our proposed algorithms for multi-label classification. All the methods are implemented in Matlab and all experiments are conducted on a workstation with a 3.2GHZ Intel CPU and 4GB main memory running 64-bit Windows platform.

## 5.1 Data sets and baselines

We conduct experiments on eight real-world data sets with various domains from three websites.[345] Following the experimental settings in [5] and [7], we preprocess the LLog, yahoo_art, eurlex_sm and eurlex_ed data sets. Their statistics are presented in the Supplementary Materials. We compare our algorithms with some baseline methods: BR, CC, ECC, CCA [14] and MMOC [7]. To perform a fair comparison, we use the same linear classification/regression package LIBLINEAR [21] with L2-regularized square hinge loss (primal) to train the classifiers for all the methods. ECC is averaged over several CC predictions with random order and the ensemble size in ECC is set to 10 according to [5, 6]. In our experiment, the running time of PCC and EPCC [5] on most data sets, like slashdot and yahoo_art, takes more than one week. From the results in [5], ECC is comparable with EPCC and outperforms PCC, so we do not consider PCC and EPCC here. CCA and MMOC are two state-of-the-art encoding-decoding [13] methods. We cannot get the results of CCA and MMOC on yahoo_art_10, eurlex_sm_10 and eurlex_ed_10 data sets in one week. Following [22], we consider the Example-F1, Macro-F1 and Micro-F1 measures to evaluate the prediction performance of all methods. We perform 5-fold cross-validation on each data set and report the mean and standard error of each evaluation measurement. The running time complexity comparison is reported in the Supplementary Materials.

Table 1: Results of Example-F1 on the various data sets (mean $\pm$ standard deviation). The best results are in bold. Numbers in square brackets indicate the rank.

| Data set | BR | CC | ECC | CCA | MMOC | CC-Greedy | CC-DP |
|---|---|---|---|---|---|---|---|
| yeast | 0.6076 ± 0.019[6] | 0.5850± 0.033[7] | 0.6096± 0.018[5] | 0.6109 ± 0.024[4] | 0.6132 ± 0.021 [3] | **0.6144**± 0.021[1] | 0.6135± 0.015[2] |
| image | 0.5247 ± 0.025[7] | **0.5991**± 0.021[1] | 0.5947± 0.015[4] | 0.5947± 0.009[4] | 0.5960± 0.012[3] | 0.5939± 0.021[6] | 0.5976± 0.015[2] |
| slashdot | 0.4898 ± 0.024[6] | 0.5246± 0.028[4] | 0.5123± 0.027[5] | 0.5260± 0.021[3] | 0.4895 ± 0.022[7] | 0.5266± 0.022[2] | **0.5268**± 0.022[1] |
| enron | 0.4792 ± 0.017[7] | 0.4799± 0.011[6] | 0.4848± 0.014[4] | 0.4812± 0.024[5] | **0.4940** ± 0.016[1] | 0.4894 ± 0.016[2] | 0.4880± 0.015[3] |
| LLog_10 | 0.3138 ± 0.022[6] | 0.3219± 0.028[4] | 0.3223± 0.030[3] | 0.2978 ± 0.026[7] | 0.3153 ± 0.026[5] | 0.3269± 0.023[2] | **0.3298**± 0.025[1] |
| yahoo_art_10 | 0.4840 ± 0.023[5] | 0.5013± 0.022[4] | 0.5070± 0.020[3] | - | - | 0.5131± 0.015[2] | **0.5135**± 0.020[1] |
| eurlex_sm_10 | 0.8594 ± 0.003[5] | **0.8609**± 0.004[1] | 0.8606± 0.003[3] | - | - | 0.8600± 0.004[4] | **0.8609**± 0.004[1] |
| eurlex_ed_10 | 0.7170 ± 0.012[5] | 0.7176± 0.012[4] | 0.7183± 0.013[2] | - | - | 0.7183± 0.013[2] | **0.7190**± 0.013[1] |
| Average Rank | 5.88 | 3.88 | 3.63 | 4.60 | 3.80 | 2.63 | 1.50 |

## 5.2 Prediction performance

Example-F1 results for our method and baseline approaches in respect of the different data sets are reported in Table 1. Other measure results are reported in the Supplementary Materials. From the results, we can see that: 1) BR is much inferior to other methods in terms of Example-F1. Our experiment provides empirical evidence that the label correlations exist in many real word data sets and because BR ignores the information about the correlations between the labels, BR achieves poor performance on most data sets. 2) CC improves the performance of BR, however, it underperforms ECC. This result verifies the answer to our first question stated in Section 1: the label order does affect the performance of CC; ECC, which averages over several CC predictions with random order, improves the performance of CC. 3) CC-DP and CC-Greedy outperforms CCA and MMOC. This studies verify that optimal CC achieve competitive results compared with state-of-the-art encoding-decoding approaches. 4) Our proposed CC-DP and CC-Greedy algorithms are successful on most data sets. This empirical result also verifies the answers to the last two questions stated in Section 1: the globally optimal CC exists and CC-DP can find the globally optimal CC which achieves the best prediction performance; the CC-Greedy algorithm achieves comparable prediction performance with CC-DP, while it requires lower time complexity than CC-DP. In the experiment, our proposed algorithms are much faster than CCA and MMOC in terms of both training and testing time, and achieve the same testing time with CC. Through the training time for our algorithms is slower than BR, CC and ECC. Our extensive empirical studies show that our algorithms achieve superior performance than those baselines.

# 6 Conclusion

To improve the performance of multi-label classification, a plethora of models have been developed to capture label correlations. Amongst them, classifier chain is one of the most popular approaches due to its simplicity and good prediction performance. Instead of proposing a new learning model, we discuss three important questions in this work regarding the optimal classifier chain stated in Section 1. To answer these questions, we first propose a generalized CC model. We then provide a theoretical analysis of the generalization error for the proposed generalized model. Based on our results, we obtain the answer to the second question: the globally optimal CC exists only if the minimization of the upper bound is achieved over this CC. It is very expensive to search over $q!$ different label orders to find the globally optimal CC. Thus, we propose the CC-DP algorithm to simplify the search algorithm, which requires $\mathcal{O}(q^3nd)$ complexity. To speed up the CC-DP algorithm, we propose a CC-Greedy algorithm to find a locally optimal CC, where the time complexity of the CC-Greedy algorithm is $\mathcal{O}(q^2nd)$. Comprehensive experiments on eight real-world multi-label data sets from different domains verify our theoretical studies and the effectiveness of proposed algorithms.

**Acknowledgments**

This research was supported by the Australian Research Council Future Fellowship FT130100746.

## Footnotes

[1] ! represents the factorial notation.

[2] The expression $[\![y_t \neq h(x_t)]\!]$ evaluates to 1 if $y_t \neq h(x_t)$ is true and to 0 otherwise.

[3]http://mulan.sourceforge.net

[4]http://meka.sourceforge.net/#datasets

[5]http://cse.seu.edu.cn/people/zhangml/Resources.htm#data

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
