[Supplementary Material]

# On the Optimality of Classifier Chain for Multi-label Classification (Supplementary)

**Weiwei Liu**            **Ivor W. Tsang**[*]
Centre for Quantum Computation and Intelligent Systems
University of Technology, Sydney
liuweiwei863@gmail.com, ivor.tsang@uts.edu.au

## Abstract

In this supplementary file, we will first present the definition of covering numbers. Then, we provide the proof of the symmetrization lemma and correctness of CC-DP appeared in the main paper. After that, we present the details of the CC-Greedy algorithm and conduct the running time complexity analysis for all the methods used in the main paper. Finally, we present the statistics on the data sets used in the paper and some results.

## 1 Covering numbers

**Definition 1** (Covering Numbers). *Let $(X, d)$ be a (pseudo-) metric space, $A$ be a subset of $X$ and $\epsilon > 0$. A set $B \subseteq X$ is an $\epsilon$-cover for $A$, if for every $a \in A$, there exists $b \in B$ such that $d(a, b) < \epsilon$. The $\epsilon$-covering number of $A$, $\mathcal{N}_d(\epsilon, A)$, is the minimal cardinality of an $\epsilon$-cover for $A$ (if there is no such finite cover then it is defined to be $\infty$).*

In the main paper, suppose $\mathcal{N}(\epsilon, \mathcal{H}, \mathbf{s})$ be the $\epsilon$-covering number of $\mathcal{H}$ with respect to the $l_\infty$ pseudo-metric measuring the maximum discrepancy on the sample $\mathbf{s}$, that is, with respect to the distance $d(f, g) = \max_{1 \leq t \leq m} |f(x_t) - g(x_t)|$, for $f, g \in \mathcal{H}$.

## 2 Proof of Lemma 2

**Lemma 2** (Symmetrization). *Let $\mathcal{H}$ be the real valued function class. $\mathbf{s}$ and $\bar{\mathbf{s}}$ are $m$ samples both drawn independently according to the unknown distribution $D$. If $m\epsilon^2 \geq 2$, then*

$$P_{\mathbf{s}}(\sup_{h \in \mathcal{H}} |er_D[h] - er_{\mathbf{s}}[h]| \geq \epsilon) \leq 2P_{\mathbf{s}\bar{\mathbf{s}}}(\sup_{h \in \mathcal{H}} |er_{\bar{\mathbf{s}}}[h] - er_{\mathbf{s}}[h]| \geq \epsilon/2) \tag{1}$$

*Proof.* (of Lemma 2). For each $\mathbf{s}$, let $\bar{h}_{\mathbf{s}}$ be a function for which $|er_D[\bar{h}_{\mathbf{s}}] - er_{\mathbf{s}}[\bar{h}_{\mathbf{s}}]| \geq \epsilon$ if such a function exists, and any fixed function in $\mathcal{H}$ otherwise. Then

$$P_{\mathbf{s}\bar{\mathbf{s}}}(\sup_{h \in \mathcal{H}} |er_{\bar{\mathbf{s}}}[h] - er_{\mathbf{s}}[h]| \geq \epsilon/2) \geq P_{\mathbf{s}\bar{\mathbf{s}}}(|er_{\bar{\mathbf{s}}}[\bar{h}_{\mathbf{s}}] - er_{\mathbf{s}}[\bar{h}_{\mathbf{s}}]| \geq \epsilon/2)$$

$$\geq P_{\mathbf{s}\bar{\mathbf{s}}}(\{|er_D[\bar{h}_{\mathbf{s}}] - er_{\mathbf{s}}[\bar{h}_{\mathbf{s}}]| \geq \epsilon\} \bigcap \{|er_{\bar{\mathbf{s}}}[\bar{h}_{\mathbf{s}}] - er_D[\bar{h}_{\mathbf{s}}]| \leq \epsilon/2\})$$

$$= E_{\mathbf{s}}[\![(|er_D[\bar{h}_{\mathbf{s}}] - er_{\mathbf{s}}[\bar{h}_{\mathbf{s}}]| \geq \epsilon)]\!] P_{\bar{\mathbf{s}}}(|er_{\bar{\mathbf{s}}}[\bar{h}_{\mathbf{s}}] - er_D[\bar{h}_{\mathbf{s}}]| \leq \epsilon/2)] \tag{2}$$

Now the conditional probability inside can be bounded using Chebyshev's inequality:

$$P_{\bar{\mathbf{s}}}(|er_{\bar{\mathbf{s}}}[\bar{h}_{\mathbf{s}}] - er_D[\bar{h}_{\mathbf{s}}]| \leq \epsilon/2) \geq 1 - \frac{\mathbf{Var}_{\bar{\mathbf{s}}}[er_{\bar{\mathbf{s}}}[\bar{h}_{\mathbf{s}}]]}{\epsilon^2/4} \tag{3}$$

---

[*]Corresponding author

Since $\bar{\mathbf{s}} \sim D^m$ and $er_{\bar{\mathbf{s}}}[\bar{h}_{\mathbf{s}}]$ is $1/m$ times a Binomial random variable with parameters $(m, er_D[\bar{h}_{\mathbf{s}}])$, we have $\mathbf{Var}_{\bar{\mathbf{s}}}[er_{\bar{\mathbf{s}}}[\bar{h}_{\mathbf{s}}]] = \frac{er_D[\bar{h}_{\mathbf{s}}](1-er_D[\bar{h}_{\mathbf{s}}])}{m} \leq \frac{1}{4m}$. This gives

$$P_{\bar{\mathbf{s}}}(|er_{\bar{\mathbf{s}}}[\bar{h}_{\mathbf{s}}] - er_D[\bar{h}_{\mathbf{s}}]| \leq \epsilon/2) \geq 1 - \frac{1}{m\epsilon^2} \geq \frac{1}{2} \tag{4}$$

whenever $m\epsilon^2 \geq 2$. Thus we get

$$\begin{aligned}
P_{\mathbf{s}\bar{\mathbf{s}}}(\sup_{h \in \mathcal{H}} |er_{\bar{\mathbf{s}}}[h] - er_{\mathbf{s}}[h]| \geq \epsilon/2) &\geq \frac{1}{2} P_{\mathbf{s}}(|er_D[\bar{h}_{\mathbf{s}}] - er_{\mathbf{s}}[\bar{h}_{\mathbf{s}}]| \geq \epsilon) \\
&= \frac{1}{2} P_{\mathbf{s}}(\sup_{h \in \mathcal{H}} |er_D[h] - er_{\mathbf{s}}[h]| \geq \epsilon)
\end{aligned} \tag{5}$$

where the last step of Eq. (5) is by definition of $\bar{h}_{\mathbf{s}}$. $\qquad\square$

## 3 Proof of Theorem 3

Recall that in the main paper, given label set $\mathcal{M} = \{\lambda_1, \lambda_2, \cdots, \lambda_q\}$, suppose a GCC model contains $q$ classifiers. Let $o_i(1 \leq o_i \leq q)$ denote the order of $\lambda_i$ is the GCC model, $\gamma_i^{o_i}$ represent the margin for label $\lambda_i$, with previous $o_i - 1$ labels as the augmented input. If $o_i = 1$, then $\gamma_i^1$ represent the margin for label $\lambda_i$, without augmented input. Then $Q'$ is defined as $Q' = \sum_{i=1}^{q} \frac{1}{(\gamma_i^{o_i})^2}$. We propose CC-DP algorithm in the main paper to find the globally optimal label order, which can minimize $Q'$. Assume $i \in \{1, \cdots, q\}$. Let $V(i, \eta)$ be the optimal $Q'$ over a subset of $\mathcal{M}$ with the length of $\eta(1 \leq \eta \leq q)$, where the label order is ending by label $\lambda_i$. Suppose $M_i^{\eta}$ represent the corresponding label set for $V(i, \eta)$. When $\eta = q$, $V(i, q)$ be the optimal $Q'$ over $\mathcal{M}$, where the label order is ending by label $\lambda_i$. The DP Equation can be written as:

$$V(i, \eta+1) = \min_{j \neq i, \lambda_i \notin M_j^{\eta}} \{\frac{1}{(\gamma_i^{\eta+1})^2} + V(j, \eta)\} \tag{6}$$

where $\gamma_i^{\eta+1}$ is the margin for label $\lambda_i$, with $M_j^{\eta}$ as the augmented input. The initial condition of DP is: $V(i, 1) = \frac{1}{(\gamma_i^1)^2}$ and $M_i^1 = \{\lambda_i\}$. Then, the optimal $Q'$ over $\mathcal{M}$ can be obtained by solving $\min_{i \in \{1, \cdots, q\}} V(i, q)$. Assume that the training of linear SVM takes $\mathcal{O}(nd)$ time complexity, we iteratively solve this DP Equation to get the optimal solution, which requires at most $\mathcal{O}(q^3 nd)$ time complexity and we have the following theorem:

**Theorem 3** (Correctness of CC-DP). *$Q'$ can be minimized by CC-DP, which means this Algorithm can find the globally optimal CC.*

*Proof.* (of Theorem 3). We proof the theorem using the mathematical induction. For $i \in \{1, \cdots, q\}$,

**Case 1:** $V(i, 1) = \frac{1}{(\gamma_i^1)^2}$, where $\gamma_i^1$ is the margin for label $\lambda_i$, without augmented input and $M_i^1 = \{\lambda_i\}$.

**Case 2:** $V(i, 2) = \min_{j \neq i, \lambda_i \notin M_j^1} \{(\frac{1}{(\gamma_i^2)^2} + V(j, 1)\}$, where $\gamma_i^2$ is the margin for label $\lambda_i$, with $M_j^1$ as the augmented input. As in case 1, we already calculated $V(i, 1)$, so we can easily find the solution of $V(i, 2)$. Assume $V(j, 1)$ is the optimal value for computing $V(i, 2)$, then we can get $M_i^2 = M_j^1 \cup \{\lambda_i\}$.

**Case 3:** Assume $V(i, k-1), k \leq q$ is the optimal $Q'$ over a subset of $\mathcal{M}$ with the length of $k-1$, where the label order is ending by label $\lambda_i$ and $M_i^{k-1}$ denote the corresponding label set for $V(i, k-1)$.

**Case 4:** $V(i, k) = \min_{j \neq i, \lambda_i \notin M_j^{k-1}} \{\frac{1}{(\gamma_i^k)^2} + V(j, k-1)\}$, where $\gamma_i^k$ is the margin for label $\lambda_i$, with $M_j^{k-1}$ as the augmented input. Based on the assumption in case 3, we can obtain $V(i, k), i \in \{1, \cdots, q\}$. Thus, we can find the optimal $Q'$ over $\mathcal{M}$ by using CC-DP algorithm. $\qquad\square$

Table 1: Time complexity.

| Method | Training time complexity | Testing time complexity |
|---|---|---|
| BR,CC,ECC | $\mathcal{O}(ndq)$ | $\mathcal{O}(dq)$ |
| CCA | $\mathcal{O}(\psi^3 + n(d^2 + q^2 + dq))$ | $\mathcal{O}(q^3)$ |
| MMOC | $\mathcal{O}(nq^3 + nq^2d + n^4)$ | $\mathcal{O}(q^3)$ |
| CC-Greedy | $\mathcal{O}(q^2nd)$ | $\mathcal{O}(dq)$ |
| CC-DP | $\mathcal{O}(q^3nd)$ | $\mathcal{O}(dq)$ |

## 4 CC-Greedy algorithm

To speed up the CC-DP algorithm, we propose a CC-Greedy algorithm to find a locally optimal CC.

Based on the training instances, we select the label from $\{\lambda_1, \lambda_2, \cdots, \lambda_q\}$ as the first label, the maximum margin can be achieved over this label, without augmented input. The first label is denoted by $\zeta_1$. Then, we select the label from the remainder as the second label, if the maximum margin can be achieved over this label with $\zeta_1$ as the augmented input. We continue in this way until the last label is selected. Finally, this algorithm will converge to the locally optimal CC, which requires at most $\mathcal{O}(q^2nd)$ time complexity. This section present the details of CC-Greedy algorithm:

---

**Algorithm 1** Greedy algorithm for locally optimal CC (CC-Greedy)

---

**Input:** training data $\{\mathbf{x}_t, \mathbf{y}_t\}_{t=1}^n$ with size $n$ and label set $\{\lambda_1, \lambda_2, \cdots, \lambda_q\}$.
Set $\mathcal{M} = \{\lambda_1, \lambda_2, \cdots, \lambda_q\}$.
**for** $\lambda_j \in \mathcal{M}$ **do**
    Calculate $[\mathbf{w}_j, b] = SVM(\{\mathbf{x}_t\}_{t=1}^n, \{\mathbf{y}_t(\lambda_j)\}_{t=1}^n)$.
    Calculate $\gamma_j^1$ using Eq. (1) in the main paper.
**end for**
Calculate $\nu = \arg_{\lambda_j \in \mathcal{M}} \min \frac{1}{(\gamma_j^1)^2}$.
Set $\mathcal{M} = \mathcal{M} - \{\lambda_\nu\}$
Set $Q[1] = \frac{1}{(\gamma_\nu^1)^2}$.
Set $C[1] = \lambda_\nu$.
**for** $s = 2$ **to** $q$ **do**
    **for** $\lambda_k \in \mathcal{M}$ **do**
        Calculate $[\mathbf{w}_k, b] = SVM(\{\mathbf{x}_t, \mathbf{y}_t(C[1]), \cdots, \mathbf{y}_t(C[s-1])\}_{t=1}^n, \{\mathbf{y}_t(\lambda_k)\}_{t=1}^n)$.
        Calculate $\gamma_k^s$ using Eq. (1) in the main paper.
    **end for**
    Calculate $\nu = \arg_{\lambda_k \in \mathcal{M}} \min \frac{1}{(\gamma_k^s)^2}$ .
    Set $\mathcal{M} = \mathcal{M} - \{\lambda_\nu\}$.
    Set $Q[s] = Q[s-1] + \frac{1}{(\gamma_\nu^s)^2}$.
    Set $C[s] = \lambda_\nu$.
**end for**
Output this locally optimal CC.

---

## 5 Complexity analysis

Assume that the training of linear SVM takes $\mathcal{O}(nd)$ time complexity. Following the running time analysis in [1], assume $q < d$, CC will takes $\mathcal{O}(ndq)$ time complexity. Let $\psi = \max\{n, d\}$. Table 1 reports the training and testing time complexity of the methods used in the main paper. From Table 1, we can see that our proposed algorithms are much faster than CCA and MMOC in terms of both training and testing time complexity, and achieve the same testing time complexity with BR, CC and ECC. Through the training time for our algorithms is slower than BR, CC and ECC. Our extensive empirical studies demonstrate that our algorithms achieve superior performance than those baselines.

Table 2: Data sets used in the experiments.

| Data | # inst. | # attr. | # labels | Domain |
|------|---------|---------|----------|--------|
| yeast | 2,417 | 103 | 14 | biology |
| image | 2,000 | 294 | 5 | image |
| slashdot | 3,782 | 1,079 | 22 | text |
| enron | 1,702 | 1,001 | 53 | text |
| LLog_10 | 799 | 1,004 | 10 | linguistics |
| yahoo_art_10 | 6,849 | 23,146 | 10 | art |
| eurlex_sm_10 | 11,454 | 5,000 | 10 | text |
| eurlex_ed_10 | 6,540 | 5,000 | 10 | text |

# 6 Data sets and results

In this section, we will report the statistics on the data sets used in the main paper and some experiment results.

## 6.1 Data sets

We conduct experiments on eight real-world data sets with various domains from three different websites.[1][2][3] Following the experimental settings in [2] and [3], we preprocess the LLog, yahoo_art, eurlex_sm and eurlex_ed data sets. The statistics on those data sets are presented in Table 2.

## 6.2 Macro-F1 and Micro-F1 results

We consider the following evaluation measurements [4] to measure the prediction performance of all methods fairly:

- Example-F1: computes the F-1 score for all the labels of each testing sample and then takes the average of the F-1 score.
- Macro-F1: calculates the F-1 score for each label and then takes the average of the F-1 score.
- Micro-F1: computes true positives, true negatives, false positives and false negatives over all labels, and then calculates an overall F-1 score.

The larger the value of those measurements, the better the performance. We perform 5-fold cross-validation on each data set and report the mean and standard error of each evaluation measurement.

The results of Macro-F1 and Micro-F1 for our method and baseline approaches in respect of the different data sets are reported in Tables 3 and 4. From the results, we can see that:

- BR generally underperforms in terms of Macro-F1 and Micro-F1.
- CC and ECC improves the performance of BR. They outperform CCA and MMOC in terms of Macro-F1, however, they underperform CCA and MMOC in terms of Micro-F1. This means CC and ECC are sensitive to the measurements.
- CC-DP outperforms CCA and MMOC stably. This studies verify that optimal CC achieve competitive results compared with state-of-the-art encoding-decoding approaches.
- CC-Greedy and CC-DP achieve more accurate performance than CC and ECC. This empirical result also verifies the answers to the last two questions stated in the main paper: the globally optimal CC exists and CC-DP can find the globally optimal CC which achieves the best prediction performance; the CC-Greedy algorithm achieves comparable prediction performance with CC-DP, while it requires lower time complexity than CC-DP.

Table 3: Results of Macro-F1 on the various data sets (mean $\pm$ standard deviation). The best results are in bold. Numbers in square brackets indicate the rank.

| Data set | BR | CC | ECC | CCA | MMOC | CC-Greedy | CC-DP |
|---|---|---|---|---|---|---|---|
| yeast | 0.3543 $\pm$ 0.014[4] | **0.3993**$\pm$ 0.027[1] | 0.3763$\pm$ 0.015[2] | 0.3496 $\pm$ 0.017[5] | 0.3431 $\pm$ 0.016[7] | 0.3441$\pm$ 0.016[6] | 0.3596$\pm$ 0.020[3] |
| image | 0.5852 $\pm$ 0.012[7] | **0.6013**$\pm$ 0.018[1] | 0.5988$\pm$ 0.010[4] | 0.6010 $\pm$ 0.009[2] | 0.5975 $\pm$ 0.007[6] | 0.5987$\pm$ 0.019[5] | 0.6010$\pm$ 0.014[2] |
| slashdot | 0.3416 $\pm$ 0.014[4] | 0.3485$\pm$ 0.015[2] | 0.3331$\pm$ 0.011[7] | **0.3512** $\pm$ 0.018[1] | 0.3334 $\pm$ 0.009[6] | 0.3431$\pm$ 0.010[3] | 0.3408$\pm$ 0.008[5] |
| enron | 0.2089 $\pm$ 0.024[2] | 0.2066$\pm$ 0.022[5] | 0.2088$\pm$ 0.022[3] | 0.1594 $\pm$ 0.027[6] | 0.1539 $\pm$ 0.017[7] | **0.2090**$\pm$ 0.024[1] | 0.2082$\pm$ 0.022[4] |
| LLog_10 | 0.3452 $\pm$ 0.030[2] | 0.3428$\pm$ 0.033[4] | 0.3425$\pm$ 0.039[5] | 0.3189 $\pm$ 0.035[7] | 0.3303 $\pm$ 0.040[6] | 0.3448$\pm$ 0.032[3] | **0.3471**$\pm$ 0.035[1] |
| yahoo_art_10 | 0.4836 $\pm$ 0.014[4] | 0.4816$\pm$ 0.013[5] | 0.4851$\pm$ 0.015[3] | - | - | 0.4876$\pm$ 0.012[2] | **0.4884**$\pm$ 0.015[1] |
| eurlex_sm_10 | 0.8546 $\pm$ 0.002[5] | 0.8558$\pm$ 0.002[2] | 0.8554$\pm$ 0.002[3] | - | - | 0.8550$\pm$ 0.002[4] | **0.8559**$\pm$ 0.003[1] |
| eurlex_ed_10 | 0.7201 $\pm$ 0.008[5] | 0.7202$\pm$ 0.008[4] | 0.7205$\pm$ 0.009[3] | - | - | 0.7208$\pm$ 0.009[2] | **0.7217**$\pm$ 0.008[1] |
| Average Rank | 4.13 | 3.00 | 3.75 | 4.20 | 6.40 | 3.25 | 2.25 |

Table 4: Results of Micro-F1 on the various data sets (mean $\pm$ standard deviation). The best results are in bold. Numbers in square brackets indicate the rank.

| Data set | BR | CC | ECC | CCA | MMOC | CC-Greedy | CC-DP |
|---|---|---|---|---|---|---|---|
| yeast | 0.6320 $\pm$ 0.019[4] | 0.6185$\pm$ 0.029[7] | 0.6306$\pm$ 0.017[5] | **0.6362** $\pm$ 0.025[1] | 0.6361 $\pm$ 0.021[2] | 0.6303$\pm$ 0.022[6] | 0.6328$\pm$ 0.017[3] |
| image | 0.5840 $\pm$ 0.015[7] | 0.5994$\pm$ 0.017[2] | 0.5955$\pm$ 0.012[5] | **0.6003** $\pm$ 0.010[1] | 0.5958$\pm$ 0.011[4] | 0.5946$\pm$ 0.019[6] | 0.5980$\pm$ 0.013[3] |
| slashdot | 0.5233 $\pm$ 0.024[6] | 0.5278$\pm$ 0.027[3] | 0.5175$\pm$ 0.025[7] | **0.5844** $\pm$ 0.022[1] | 0.5720 $\pm$ 0.022[2] | 0.5266$\pm$ 0.023[5] | 0.5272$\pm$ 0.023[4] |
| enron | 0.5052 $\pm$ 0.013[6] | 0.5013$\pm$ 0.009[7] | 0.5056$\pm$ 0.010[5] | 0.5335 $\pm$ 0.015[2] | **0.5401** $\pm$ 0.010[1] | 0.5104$\pm$ 0.013[3] | 0.5096$\pm$ 0.012[4] |
| LLog_10 | **0.3768** $\pm$ 0.028[1] | 0.3712$\pm$ 0.030[6] | 0.3730$\pm$ 0.035[5] | 0.3623 $\pm$ 0.027[7] | 0.3760 $\pm$ 0.027[3] | 0.3744$\pm$ 0.028[4] | 0.3762$\pm$ 0.029[2] |
| yahoo_art_10 | 0.5122 $\pm$ 0.017[5] | 0.5130$\pm$ 0.016[4] | 0.5156$\pm$ 0.018[3] | - | - | **0.5184**$\pm$ 0.013[1] | **0.5184**$\pm$ 0.017[1] |
| eurlex_sm_10 | 0.8718 $\pm$ 0.001[5] | 0.8727$\pm$ 0.001[2] | 0.8725$\pm$ 0.001[3] | - | - | 0.8722$\pm$ 0.001[4] | **0.8733**$\pm$ 0.002[1] |
| eurlex_ed_10 | 0.7419 $\pm$ 0.009[5] | 0.7421$\pm$ 0.009[4] | 0.7424$\pm$ 0.010[3] | - | - | 0.7425$\pm$ 0.010[2] | **0.7432**$\pm$ 0.010[1] |
| Average Rank | 4.88 | 4.38 | 4.50 | 2.40 | 2.40 | 3.88 | 2.38 |

## Footnotes

[1]http://mulan.sourceforge.net

[2]http://meka.sourceforge.net/#datasets

[3]http://cse.seu.edu.cn/people/zhangml/Resources.htm#data