[Reviews · NeurIPS 2015]

Submitted by Assigned_Reviewer_1

The authors have developed a new algorithm for multi-label classification. They have supplied theoretical analysis. They have also supplied empirical analysis in which they have improved the state of the art. The paper is well motivated and clearly written. The contents of the paper is quite original. Further the proposed algorithm is not theoretically interesting but also practically interesting.

In the future, I would like to see the results of the method on large scale datasets.
Summary: The authors have developed a new algorithm for multi-label classification. They have supplied theoretical analysis. They have also supplied empirical analysis in which they have improved the state of the art. I suggest acceptance of the paper.

Submitted by Assigned_Reviewer_2

Quality The paper appears to be of high quality and contains a substantial theoretical advance.

Clarity The paper presentation is clear.

Originality The work is original.

Significance This work is likely of significance to the community.
Summary: The paper considers the problem of impact of label order in classifier chain (CC) multi-label classification and proposes two algorithms CC-DP and CC-Greedy which in effect optimize this order. In extensive experiments, the algorithms appear to outperform other state of the art algorithms. The paper significantly contributes to understanding the impact of label order in the CC approach.

Submitted by Assigned_Reviewer_3

The paper deals with the construction of optimal classifier chains for multi-label classification. The first part of the paper provides a bound on the error of the classifer chain in terms of the margins over the labels. This result is used in the second part to derive a DP and greedy algorithm. The experimental section shows that that the DP approach has the best results in 5 out of 8 datasets and the greedy approach in 1.

Quality: The bound on error involves interesting ideas. Hence the paper scores high on quality.

Clarity: The exposition is clear. The authors should address the following questions.

1. The greedy approach performs better than DP on 2 out of 8 datasets. Some explanation will be helpful. 2. Comparison of actual runtimes of greedy vs DP will help.

The paper has some typos. For example, line 313 should read "\gamma^1_i represents ..."

Orginality: High

Significance: The techniques used in the paper have theoretical significance. I have not seen many people using multilabel classification in practice.
Summary: The paper has strong theoretical content which is also empirically validated.

Author Feedback
Author rebuttal: We thank the reviewers for their complimentary comments and will address all the reviewer's comments in the updated manuscript. Next, we answer the following questions raised here.

Q: The "generalized" classifier chain model proposed in this paper....

A: Following [5] and [6], CC does not use the random label order. To clearly state the analysis, we generalize the CC model over a random label order.

Q: ... the results of PCC and EPCC ....

A: In our experiment, the running time of PCC and EPCC on most data sets, like slashdot and yahoo_art, takes more than one week. From the results in [5], ECC is comparable with EPCC and outperforms PCC, so we do not consider PCC and EPCC in the paper.